# Sirtuins and Renal Oxidative Stress

**DOI:** 10.3390/antiox10081198

**Published:** 2021-07-27

**Authors:** Yoshio Ogura, Munehiro Kitada, Daisuke Koya

**Affiliations:** 1Department of Diabetology and Endocrinology, Kanazawa Medical University, Uchinada, Ishikawa 920-0293, Japan; kitta@kanazawa-med.ac.jp (M.K.); koya0516@kanazawa-med.ac.jp (D.K.); 2Division of Anticipatory Molecular Food Science and Technology, Medical Research Institute, Kanazawa Medical University, Uchinada, Ishikawa 920-0293, Japan

**Keywords:** chronic kidney disease (CKD), oxidative stress, sirtuins

## Abstract

Renal failure is a major health problem that is increasing worldwide. To improve clinical outcomes, we need to understand the basic mechanisms of kidney disease. Aging is a risk factor for the development and progression of kidney disease. Cells develop an imbalance of oxidants and antioxidants as they age, resulting in oxidative stress and the development of kidney damage. Calorie restriction (CR) is recognized as a dietary approach that promotes longevity, reduces oxidative stress, and delays the onset of age-related diseases. Sirtuins, a type of nicotinamide adenine dinucleotide (NAD)-dependent histone deacetylase, are considered to be anti-aging molecules, and CR induces their expression. The sirtuin family consists of seven enzymes (Sirt1–7) that are involved in processes and functions related to antioxidant and oxidative stress, such as DNA damage repair and metabolism through histone and protein deacetylation. In fact, a role for sirtuins in the regulation of antioxidants and redox substances has been suggested. Therefore, the activation of sirtuins in the kidney may represent a novel therapeutic strategy to enhancing resistance to many causative factors in kidney disease through the reduction of oxidative stress. In this review, we discuss the relationship between sirtuins and oxidative stress in renal disease.

## 1. Kidney and Oxidative Stress

Renal failure is a major health problem that is increasing worldwide. There are two types of kidney disease: acute kidney injury (AKI) and chronic kidney disease (CKD). AKI is reversible, but it can also progress to end-stage renal disease (ESRD). CKD, however, is persistently chronic and increases the risk of death by two fold [1]. CKD is recognized as a global health problem due to the high cost of medical care. There are various pathological mechanisms involved in CKD. These include altered signaling pathways, mitochondrial dysfunction and oxidative stress, altered autophagy, chronic inflammation, vascular dysfunction, and epigenetic regulation [2,3]. Oxidative stress plays a pivotal role in these CKD pathologies. Since kidneys are the most susceptible organs to hypoxia and renal hypoxia, which usually occurs during the course of renal disease progression, hypoxia is thought to play an important role in exacerbating oxidative stress in CKD. In general, renal hypoxia is a prognostic factor for various renal diseases and is an important issue that needs to be addressed [4]. Increased renal oxidative stress leads to chronic tubulointerstitial hypoxia and increased tubular destruction due to stagnant blood flow in the tubular capillaries [4]. Chronic tubulointerstitial hypoxia not only occurs in parallel with the progression of CKD, but is also considered to be a common final step in the development of ESRD, which has various causes [5]. In other words, renal hypoxia is both a cause and a consequence of kidney disease, and this vicious cycle contributes to the progression of CKD [6]. In fact, in the last few years, several experimental studies have shown that oxidative imbalance is common in CKD [7,8]. Oxidative stress is one of the key factors that trigger the inflammatory process [9].

Typical CKD patients suffer from chronic inflammation, and chronic inflammation and oxidative stress, if not properly controlled, can cause a variety of adverse effects, including overproduction of cytokine [9]. Furthermore, oxidative stress is thought to be particularly enhanced in patients with diabetic kidney disease (DKD) [10,11]. In diabetes mellitus, chronic hyperglycemia (HG) induces oxidative stress, and oxidative stress-mediated renal inflammation and fibrosis contribute to the pathogenesis of DKD [12].

Obesity is also a strong risk factor for CKD because obese people are more likely to develop diabetes and hypertension [10,13]. It has been shown that obesity can lead to renal dysfunction and kidney damage through the direct involvement of the endocrine activity of adipose tissue. In fact, many adipokines that are produced by higher-order visceral adipose tissue have been implicated in insulin resistance, inflammation, and oxidative stress. Furthermore, oxidative stress is thought to be particularly enhanced in patients with diabetic kidney disease (DKD) [10,11].

However, persistent oxidative stress also plays an important role in the pathogenesis of AKI and is known to be involved in the transition from AKI to CKD. Recent studies have revealed that tubulointerstitial hypoxia plays a key role in the transition from AKI to CKD and the pathophysiology of CKD [14,15]. The capillary rarefaction phenomenon that occurs after AKI causes renal hypoxia, which affects tubular epithelial cells, fibroblasts, and inflammatory cells and induces tubulointerstitial fibrosis. This is the most common cause of renal hypoxia. Damaged tubules are populated by epithelial cells that have failed to re-differentiate, leading to a decrease in the vascular endothelial growth factor (VEGF), which contributes to capillary rarefaction and exacerbates hypoxia, and this forms a vicious cycle [15,16].

Furthermore, it has been reported that the treatment of inflammation and oxidative stress is important in uremic syndrome. Uremic toxins, which are high-level metabolic end products, are closely associated with CKD progression and many CKD-related complications [12]. Indoxyl sulphate (IS) toxin, one of the most common uremic toxins, causes oxidative stress, which in turn increases tissue oxygen consumption and leads to hypoxia [12]. Potentiation of other uremic toxins, such as phenyl sulphate and ρ-cresyl sulphate, has been shown to decrease the sensitivity of tubular cells to oxidative stress and to reduce glutathione levels [17].

In summary, hypoxia and oxidative stress in renal tissues, caused by inflammation, obesity, HG, and uremic toxins, are intricately interrelated, and both can exacerbate the progression of CKD and AKI, thus further damaging renal tissues through the accumulation of oxidative stress. Therefore, countermeasures and treatments for hypoxia and oxidative stress need to be developed in order to break this vicious cycle and to protect the kidneys.

## 2. Regulation of Oxidative Stress

The main effect of proper redox control is a physiological buffering system that maintains electrolyte balance and excretes toxins and waste products [18]. Cells have many defense mechanisms used to maintain a fine balance between antioxidant and oxidative systems [19], but when toxins accumulate due to inappropriate biochemical reactions within the cell or certain external factors, redox homeostasis can be altered and oxidative stress can accumulate. In other words, oxidative stress is “an imbalance between oxidants and antioxidants, leading to impaired redox signaling and regulation and/or molecular damage” [20] (Figure 1).

### 2.1. Oxidative Stress Mechanisms

In vivo, various chemically reactive molecules, including oxygen, have been found to be essential for intracellular signaling and biochemical processes. The primary players in oxidative stress are reactive oxygen species (ROS) and nitrogen species (RNS), which are collectively called reactive oxygen and nitrogen species (RONS) [21]. ROS production is a physiological process [22], and moderate or low levels of ROS are important for many cell-signaling pathways, energy extraction from organic molecules, immune defense, cell division, and redox reactions [22,23]. However, when produced in excess or combined with other RONS, ROS can have serious adverse effects on cellular components [24,25].

Endogenous sources of ROS include oxygen-consuming mitochondria, peroxisomes, endoplasmic reticulum, and various intracellular organelles [26]. The major forms of these RONS in the cell are hydrogen peroxide (H_2_O_2_), superoxide anion radicals (O_2_^•−^), nitric oxide (NO), which is an RNS, and peroxides of lipids, proteins, and nucleic acids. These free radicals exhibit very different reactivities. NADPH oxidase (nicotinamide adenine dinucleotide phosphate oxidase) (NOX) is a prevalent source of O_2_^•−^ that is formed by the addition of one electron leak from the electron transport system during cellular respiration to molecular oxygen [26]. Most superoxide is converted to hydrogen peroxide (H_2_O_2_) via the mitochondria. H_2_O_2_ is a neutral molecule because it has no unpaired electrons, but it can produce the most reactive and dangerous hydroxyl radical (OH^•^) [27]. Phospholipids and proteins in cell membranes primarily activate OH^•^.

Another important factor involved in the redox equilibrium of cells is NO. In mammals, NO has three primary isoforms of nitric oxide synthase (NOS). These are endothelial NOS (eNOS), which is involved in vasodilation and vascular regulation; neuronal NOS (nNOS), which is involved in intracellular signaling; and inducible NOS (iNOS), which is activated in response to various endotoxin and cytokine signals [28].

All isoforms of NOS use arginine as a substrate and nicotinamide adenine dinucleotide (NAD) reduced with molecular oxygen and phosphate (NADPH) as a cosubstrate. When NO reacts with O_2_^•−^, a potent oxidant, peroxynitrite (ONOO^−^) is produced [29]. This compound is responsible for oxidative damage, nitrosation, and the S-nitrosylation of biomolecules, such as DNA. Nitrosyl stress via the nitrosylation of ONOO^−^ is responsible for DNA single-strand breaks and the subsequent activation of poly-ADP-ribose polymerase (PARP) [30]. Mitochondrial dysfunction, which is caused by multiple factors, such as aging, diabetes, and inflammation, is thought to be involved in the development of RONS.

In CKD, atherosclerosis caused by the accumulation of oxidative stress due to decreased NO bioavailability arising from endothelial dysfunction promotes CKD progression. In particular, reactive oxygen species, especially O_2_^•−^, cause the inactivation or deficiency of NO, which is an important antioxidant that preserves renal function by increasing renal blood flow, increasing natriuretic pressure, regulating the tubular function, and maintaining fluid and electrolyte homeostasis. NO deficiency and high concentrations of plasma O_2_^•−^ are considered to be important promoters of oxidative stress. An example of renal RONS generation is NOX, and uremic toxins are thought to be involved in the metabolism of NOX. It has also been suggested that IS activates NOX and enhances ROS [29].

In addition, there are many exogenous sources of RONS, including air and water pollution, pesticides, tobacco, alcohol, heavy metals (Fe, Cu, etc.) and transition metals (Cd, Hg, etc.), drugs, industrial solvents, food, and radiation [31]. In the body, all of these substances are thought to be metabolized into free radicals and affect organs, such as the kidney.

### 2.2. Antioxidant Defense Mechanisms

To tightly control the levels of ROS and other RONS, cells also have myriad antioxidant defenses [23]. Antioxidants exist in both endogenous and exogenous molecules. Endogenous antioxidants (those produced naturally in situ) include both enzymatic and nonenzymatic molecules. The major enzymatic scavengers are super oxide dismutase (SOD), catalase (CAT), and haem oxygenase 1 (HO-1) glutathione peroxidase (GSH-Px), which prevent the generation of oxidative chains by scavenging the molecules responsible for generating free radicals [21,32]. Under hypoxic conditions, the respiratory chain increases the production of O_2_^•−^ [33]. The increased superoxide causes SOD in the mitochondrial matrix, which is disproportionate to the superoxide of H_2_O_2_, and ultimately causes the transcription of genes, which allows the cells to respond to hypoxia [34]. CAT breaks down H_2_O_2_ into water and oxygen [35], and HO-1 breaks down haem to produce reaction products that have antioxidant properties [36]. GSH-Px converts peroxides and hydroxyl radicals. Reduced glutathione (GSH) is oxidized to glutathione disulphide and is finally reduced to GSH, in its non-toxic form, via glutathione reductase [35].

The cellular stress response refers to the ability of cells to resist stress conditions. This phenomenon is representative of an ancient and highly conserved cytoprotective mechanism [37,38]. The response to stress requires the activation of a pro-survival pathway. This pathway is under the control of protective genes called vitagenes, which produce molecules (heat shock proteins (Hsp), glutathione, and bilirubin) with antioxidant and anti-apoptotic activities [39]. Among the intracellular pathways that protect against oxidative stress, the products from vitagenes play an important role [40,41,42]. These include members of the Hsp family, such as HO-1 and Hsp72 [43]. Some vitagenes, such as HO-1, are also upregulated in a “phase II reaction” known as the electrophilic counterattack reaction, a cytoprotective reaction that protects against a variety of oxidants. The expression of these enzymes is upregulated by the master regulator of the intracellular defense response to oxidative stress and is the nuclear factor erythroid 2-related factor 2 (Nrf2)-Kerch-like ECH-associated protein 1 (KEAP1) interaction [44]. It independently senses changes in the intracellular redox status, and in response, triggers transcriptional responses, thus regulating the expression of several antioxidant and detoxification genes. Under normal conditions, KEAP1 binds to Nrf2 and causes proteasomal degradation of Nrf2 [44]. Under oxidative stress, however, the molecular structure of KEAP1 is altered, and its ability to bind Nrf2 is lost. As a result, Nrf2 accumulates and translocates to the nucleus, where it binds to the antioxidant response element (ARE) in the promoter region of the antioxidant response gene and initiates its transcription [44]. This leads to the expression of a variety of cytoprotective genes related to redox and detoxification processes [45]. Indeed, treatments are underway to improve renal function by targeting the Nrf2/KEAP pathway [46].

Nonenzymatic antioxidants, such as L-arginine, CoQ10, and uric acid (85% antioxidant capacity in plasma), have also been reported to interact with RONS in order to terminate the free radical chain [47]. However, depending on the degree of metabolic abnormalities (diabetes, obesity, and dyslipidemia) and renal failure, antioxidants can become depleted, and the antioxidant system gradually deteriorates, resulting in an uneven oxidative equilibrium in the plasma [48,49]. This has also been reported by animal and human research groups to be the first detectable event of redox damage [50,51]. The coexistence of this disturbance in the oxidative equilibrium and activation of apoptosis has been reported to be a potential mechanism for organ damage, including renal injury. It is thought that the complex relationship between metabolic abnormalities and renal damage may be represented due to disturbances in the systemic oxidative balance.

In addition, there are exogenous nonenzymatic antioxidants, many of which are supplied by food. Ascorbic acid (vitamin C) scavenges hydroxyl radicals and superoxide radicals. Tocopherols (vitamin E), selenium, zinc, acetylcysteine, and other drugs prevent lipid peroxidation of cell membranes [52]. Recent studies have focused on hydroxytyrosol (HT) as one of the main phytochemicals found in olive oil and table olives, which are commonly used in Mediterranean diets [53]. There have been an increasing number of reports noting that HT and its derivatives activate phase II reactions, thus leading to the expression of the Nrf2 antioxidant pathway. It has been established that 40–50% of HTs contain products, an aqueous extract of olive pulp, that stimulate the vitagene pathway and act on oxidative stress via the Nrf2 pathway, thus suppressing neuroinflammation and exerting neuroprotective effects [53]. Exogenous non-enzymatic antioxidants are an area that is still being studied and attracting a lot of attention.

### 2.3. Hormesis and Oxidative Stress

However, there is a phenomenon called hormesis. Hormesis is a dose–response phenomenon characterized by stimulation at low doses and inhibition at high doses [54]. Schematically, it can be described as an inverted U-shaped dose–response, or a J-shaped or U-shaped dose–response [55]. For example, many antitumor drugs that kill tumor cells at high doses may promote tumor cell growth at low doses, which is called a hormetic dose response. At toxicologically high doses, the typical endpoints measured indicate cell damage, but when the dose falls below a threshold, low-dose stimulation meets the biological performance measures, as seen in the modest increases in cognition, growth, longevity, bone density, and other endpoints of biomedical interest. However, it is more likely to be a manifestation of an adaptive response [56].

These quantitative features of the hormesis dose–response have important medical implications. Evidence is emerging that supports the role of hormesis in low and transient increases in oxidative stress. Levels of oxidation are relatively low in most normal conditions. However, in some cells, oxidative stress increases substantially when energy demand increases. For example, intense exercise significantly increases the production of the superoxide, H_2_O_2_, OH^•^, and ONOO^−^ [57]. However, the products from free radicals generated during moderate exercise have been suggested to play an important role in the hormonal effects of exercise on muscles, including changes in energy metabolism pathways, mitochondrial biogenesis, and the upregulation of the antioxidant system [57]. Cells that are exposed to moderate transient stress will be protected from more severe stress. The change in antioxidant capacity in response to oxidative stress may allow cells to adapt to potentially damaging situations [38].

In fact, mitochondrial superoxide production is thought to contribute to neuronal damage in conditions ranging from chronic intermittent cerebral hypoxia to Alzheimer’s disease [58,59]. However, it has been reported that transient effects of low concentrations of superoxide on neurons, which are converted to H_2_O_2_, can protect them from subsequent exposure to lethal levels of stress [60,61,62].

The effect of (−)deprenyl (Selegiline, Jumex, Eldepryl, Movergan), a close structural relative to phenylethylamine, has been reported as an example of hormesis in animal models. (−)Deprenyl prolonged the lifespan of several male rats, dogs, and mice and hamsters [63]. (−)Deprenyl has also been shown to decrease the lifespan of a number of species as well. Rationalizing the toxicological findings of many of these studies, they found that they are consistent with a dose–response pattern of an increased survival and increased lifespan at low doses but a decreased lifespan at higher doses. It was concluded that the life-extending effect of (−)deprenyl at low doses is an example of hormesis. One month before the end of the 27-month longevity study, follow-up evaluation of (−)deprenyl for antioxidant enzymes (CAT, SOD) showed biphasic dose–response activity of the hormone in striatal and cortical regions of the brain. These enzymatic findings are in parallel with the longevity results, suggesting a possible causal relationship between antioxidant cell activity and longevity [55,63,64].

These findings have overturned the long-held belief that mitochondrial ROS only adversely affect cell function and survival. It is now clear that, depending on the dose, mitochondrial superoxide and H_2_O_2_ play important roles in a variety of cellular functions and activate signaling pathways, such as the vitagene pathway, that promote cell survival and disease resistance, thus revealing new aspects of oxidative stress.

## 3. Renal Oxidative Stress and Anti-Aging Therapy

A decline in cellular homeostasis with aging leads to uneven oxidative equilibrium and the accumulation of systemic oxidative stress, which is greatly involved in the pathogenesis of CKD and is one of the common factors in kidney-related death [1,65]. Furthermore, in the elderly, AKI often occurs in CKD patients, and when AKI occurs in addition to CKD, the renal prognosis is said to be poor because the pathological changes associated with CKD increases renal vulnerability or an increased vulnerability to AKI, which inhibits renal repair [66].

In other words, antiaging may be a therapeutic approach to controlling renal oxidative stress and to improving renal prognosis. Calorie restriction (CR) delays aging and prolongs its lifespan in yeast, worms, flies, and rodents. In rhesus monkeys and humans, CR inhibits age-related diseases, including metabolic diseases, CKD, and cancer, by attenuating metabolic changes and reducing inflammation and oxidative stress [67]. In fact, many studies have shown that dietary guidance/CR has a positive impact on oxidative stress by decreasing ROS levels and increasing the activity of antioxidant enzymes [68].

## 4. Kidney and Sirtuins

In early studies on yeast aging, silent information regulator 2 (Sir2), a nicotinamide adenine dinucleotide (NAD^+^)-dependent deacetylase, was identified as one of the molecules through which CR prolongs lifespan [69]. The homologue of Sir2 in higher eukaryotes is called Sirt1, and it may contribute to CR-induced lifespan extension. Sirtuin genes have attracted attention due to their association with health benefits [70]. Currently, seven sirtuins, including Sirt1, have been identified in mammals [71]. Sirt1 localizes to the nucleus and migrates to the cytoplasm under certain conditions [72]. Similarly, Sirt6 appears to localize to the cytoplasm and the nucleus [73], and Sirt2 is primarily found in the cytoplasm; Sirt3, Sirt4, and Sirt5 in the mitochondria; and Sirt7 in the nucleus and nucleolus [74]. In addition to their well-known deacetylase functions, sirtuins have evolved as mono-ADP-ribosyltransferases (Sirt4, Sirt6), lipoamidases (Sirt4) [72], and demalonylation, deglutarylation, and descutinases (Sirt5) [75] (Table 1). Sirtuins (Sirt1–7) are involved in processes and functions related to antioxidant and oxidative stress, such as mitochondrial function, DNA damage repair, and metabolism, and affect cellular energy and redox balance. Furthermore, since the deacetylation activity of sirtuins is dependent on NAD^+^, which is a redox signaling molecule, sirtuins are thought to play an important role in maintaining cellular homeostasis by regulating the expression of antioxidant enzymes and transcription factors that control the NAD^+^/NADH ratio in cells [71,76,77]. It has also been suggested that sirtuins may be a vitagene. Therefore, sirtuins may play an important role in regulating antioxidant and redox signaling pathways in cells [78]. Several of the target proteins of mammalian sirtuins have been identified as effectors of major oxidative stress pathways [79,80,81]. However, the effects of sirtuins on these effectors and the specific mechanisms of the sirtuin effects on antioxidant-induced redox signaling are very interesting areas of research. It has also been reported that sirtuins are expressed in various types of kidney cells, including glomerular cells, sarcomeric cells, mesangial cells, and tubular cells. Many previous studies on kidney disease have regarded sirtuins as age-related survival factors, but whether sirtuins are actively involved in kidney pathogenesis remains controversial. This review summarizes the possible mechanistic role of sirtuins in preventing kidney disease from an antioxidant perspective.

## 5. Sirtuins and Renal Oxidative Stress

### 5.1. Sirt1

Sirt1 is the founder and the most well-studied member of the mammalian sirtuin family. Classically thought to reside in the nucleus, several recent studies have shown that Sirt1 also localizes to the cytoplasm [82]. Sirt1 inhibits apoptosis and senescence of inflammatory and oxidative stress cells. In addition, it has been shown to regulate vascular-related factors, activation of mitochondrial biogenesis, and metabolic transcription factors involved in autophagy and nutrition [83]. Sirt1 has also been shown to preferentially deacetylate the lysine residue 9 (H3K9) of histone 3 and the lysine residue 16 (H4K16) of histone 4 [84]. There are many reports on the role of Sirt1 in oxidative stress and redox signals (Table 2).

#### 5.1.1. Sirt1 and NF-ĸB

The action of the nuclear transcription factor NF-ĸB is highly regulated by reversible acetylation, and Sirt1 has already been shown to inhibit the NF-ĸB signaling pathway by deacetylating its p65 subunit [85]. NF-ĸB is a well-known inflammatory protein that also regulates ROS and is associated with the generation of highly reactive intermediates during inflammation. Inducing NF-ĸB can further enhance the inflammatory response [86], and its levels have been shown to influence ROS levels in cells. It has also been shown that the activity of NF-kB is regulated by the levels of ROS [86]. NF-ĸB plays a dual role in regulating ROS by targeting enzymes that promote the production of ROS, such as NOX, xanthine oxidoreductase, and cytochrome p450, which is an essential component of the electron transport system, and has enzymes that are thought to contribute to the inhibition of ROS, such as SOD1 and 2, and GSH S-transferase [86].

Furthermore, it has been suggested that NF-ĸB may interact with the Nrf2/ARE pathway via microRNA-29. In HG-cultured tubular epithelial cells, reduced Sirt1 activity leads to acetylation of NF-ĸB, which directly binds to the promoter and reduces miR-29 expression, resulting in an increased expression of Keap1 and inhibition of the Nrf2/ARE pathway [87]. Another report showed that knockdown of Sirt1 inhibited the expression of antioxidant enzymes (HO-1 and SOD1) as well as Nrf2 [88]. In a previous report, microarray analysis suggested that NF-ĸB was activated in the kidneys of diabetic patients [89]. Indeed, it has been shown that p65 acetylation is enhanced in the kidneys of diabetic patients and mouse models [90]. More importantly, compared to the control diabetic model, diabetic model mice with podocyte-specific knockdown of Sirt1 exhibited increased acetylation of p65 and worsened proteinuria and renal damage. NF-ĸB has also been reported to be involved in the development of renal injury, not only in diabetic kidneys [91], but also in 5/6 nephrectomy, adenine overload nephritis, and adriamycin nephropathy [92].

As reported above, NF-ĸB is involved in the development of oxidative stress as well as inflammation and can exacerbate kidney damage; Sirt1 activation deacetylates the NF-ĸB p65 subunit. By regulating the NF-kB pathway, it may reduce oxidative stress as well as inflammation and can achieve renal protection.

#### 5.1.2. Sirt1 and p53

p53 is a tumor suppressor gene and is activated under conditions of DNA damage to induce cell cycle arrest and apoptosis. Acetylation stabilizes and activates p53 and promotes transcription of the target apoptotic genes p21 and Bax [93]. Sirt1 negatively regulates p53 activity by deacetylating the C-terminal Lys-382 residue, which inhibits p53-dependent apoptosis in response to DNA damage and oxidative stress, thus resulting in cell survival. It promotes cell survival by inhibiting p53-dependent apoptosis in response to DNA damage and oxidative stress [94]. Sirt1-dependent deacetylation of p53 regulates cell cycle progression, senescence, and apoptosis [95]. Furthermore, while p53 has traditionally been thought of as an apoptosis-regulating protein, it has been shown to play a rather complex role in the regulation of redox signaling [96]. Indeed, p53 has been shown to regulate intracellular ROS and increase certain antioxidant proteins, such as SOD2 and GSH-Px, in the absence of stress [97,98,99].

In renal tissue, there is some evidence that p53 mediates renal podocyte damage in diabetic conditions [100,101,102]. Tubular cells cultured with HG exhibited a decreased expression of Sirt1 and an increased expression of p53. Knockdown of Sirt1 with siRNA also promoted acetylation of p53 [100]. In human vascular endothelial cells, reduced Sirt1 activity increased p53 acetylation and caused a stress-induced premature aging-like phenotype, and Sirt1 activation antagonized oxidative stress-induced premature aging via the regulation of p53 [95].

These results suggest that the activation of Sirt1 may not only regulate apoptosis but may also regulate oxidative stress in a p53-dependent manner and promote the survival of renal histiocytes and vascular endothelial cells.

#### 5.1.3. Sirt1 and eNOS

Donato et al. found that downregulation of Sirt1 expression in vascular endothelial cells with aging is associated with a decrease in NO production capacity due to the deacetylation of eNOS at Lys-496 and Lys-506 [103]. Activation of Sirt1 mediates the activity of eNOS and has been reported to promote NO production and protect against premature aging caused by oxidative stress [104].

In renal tissue, renal ischemia/reperfusion (I/R) model rats displayed acute kidney injury with increased tubular cell necrosis and decreased eNOS expression, as compared to control rats. Activation of Sirt1 by CR restored the expression of eNOS in I/R model rats and had a protective effect on tubular cells [105].

As a result, it has been reported that eNOS activation by Sirt1 activation plays an important role in maintaining vascular function and homeostasis by increasing vascular endothelial cell NO production, leading to improved survival in response to ROS-induced premature senescence and cytotoxicity, as well as renal tissue protection against ischemia. The Sirt1/eNOS pathway may be a common therapeutic target for renal disease.

#### 5.1.4. Sirt1 and p66shc

The adaptor protein 66 kDa Src homology 2 domain-containing protein (p66Shc) is an important regulator of mitochondrial ROS production and mediates renal injury by promoting oxidative stress [106]. Furthermore, an elevated expression of p66Shc was associated with a significant decrease in SOD2, which is an antioxidant enzyme, suggesting that p66Shc mediates mitochondrial ROS production and leads to ROS accumulation by promoting the downregulation of scavenging enzymes. In addition, acetylation of p66Shc promotes its phosphorylation and translocation to the mitochondria, where it promotes hydrogen peroxide production [106,107]. Sirt1 negatively regulates p66Shc by promoting deacetylation of histone H3 and p66Shc [108]. According to Zhou et al., the inhibition of Sirt1 increases mRNA and protein levels of p66Shc in human embryonic kidney cells and vascular endothelial cells [109]. However, the overexpression of Sirt1 inhibits the HG-induced upregulation of p66Shc in vascular endothelial cells [109]. Furthermore, suppression of p66Shc attenuates oxidative stress and endothelial dysfunction [110]. Previous studies have shown that p66Shc-deficient mice are more resistant to oxidative stress and have a 30% longer lifespan. ROS-dependent vascular endothelial dysfunction induced by aging and HG conditions was also alleviated in these mice, which were protected from the development of certain diabetic complications, such as kidney disease [111,112]. Thus, inhibition of p66Shc maintains endothelial function, delays vascular aging, and protects against renal injury.

Furthermore, p66Shc is expressed in renal tubular cells and glomeruli, and as an adaptor protein, it plays an important role in the progression and worsening of kidney disease pathology [113]. Based on these findings, p66Shc damages renal and vascular tissues by promoting mitochondrial oxidative stress. Activation of Sirt1 reduces oxidative stress and endothelial dysfunction by negatively regulating p66Shc.

#### 5.1.5. Sirt1 and PGC-1α

Sirt1 is involved in mitochondrial function through the regulation of peroxisome proliferator-activated receptor-γ coactivator-1α (PGC-1α). PGC-1α is involved in mitochondrial biogenesis and oxidative phosphorylation proteins [114], and activation of Sirt1 promotes PGC-1 transcription and increases mitochondrial number and function. These phenomena directly affect ATP synthesis in lipid metabolism and apoptosis and are important in maintaining normal cell and organelle functioning [115].

In the kidney, Sirt1 regulates the activity of PGC-1α and plays an important role in the maintenance of mitochondrial function in podocytes [116]. Both mitochondrial damage and cellular senescence are important pathological processes that mediate kidney damage [117]. Consistent with this, a recent study demonstrated that the loss of Sirt1 in podocytes exacerbates age-related kidney disease through enhanced cellular senescence and mitochondrial dysfunction [118]. Conversely, activation of PGC-1α by increased Sirt1 activity attenuates renal fibrosis in DKD.

Thus, PGC-1α maintains mitochondrial function and reduces the stress caused by diabetes, ischemia, and aging. In this way, the Sirt1/PGC-1α pathway has been reported to affect metabolic abnormalities, which is attracting increasing attention.

#### 5.1.6. Sirt1 and AMPK

A notable aspect of Sirt1-mediated metabolic control is the regulation of AMP-activated protein kinase (AMPK), also known as the intracellular key regulator of metabolism and energy balance, and it has been reported that deacetylation of liver kinase B1 (LKB1), an AMPK upstream kinase, induces AMPK activation [119]. Conversely, activation of AMPK has been reported to increase the activity of Sirt1 by increasing NAD^+^ levels, leading to deacetylation of Sirt1 targets [120]. AMPK and Sirt1 are intracellular energy sensors that detect and respond to AMP/ATP and NAD^+^/NADH ratios, respectively [121,122]. In vitro, HG induced a decrease in the expression and activity of AMPK, Sirt1, and SOD2 in renal tubular cells [123]. In vivo, AMPK/Sirt1 activity was reduced and ROS was increased in kidney tissue in a mouse model of type 2 diabetes [124]. We previously showed that CR has a beneficial effect on DKD through the activation of AMPK and Sirt1 [125]. Aging kidneys are also characterized by reduced renal functioning, such as glomerulosclerosis, interstitial fibrosis, tubular atrophy, and vascular stiffness. Dang et al. showed that AMPK signaling in the kidney decreases with age. They also showed that short-term (8 weeks) CR may directly affect cellular senescence and pathological changes by activating the AMPK signaling pathway in aging rat kidneys [126].

These results indicate that activation of the Sirt1/AMPK pathway inhibits ROS and protects renal constituent cells, thus reducing damage in DKD and aged kidneys.

#### 5.1.7. Sirt1 and FoxO

Sirt1 deacetylates Forkhead Box O (FoxO) and alters the transcriptional activity of FoxO target genes. FoxO belongs to the Forkhead family of transcription factors with a conserved DNA binding domain. In mammals, four families have been identified: FoxO1, FoxO3a, FoxO4, and FoxO6. FoxO1 and FoxO3a are ubiquitously expressed, FoxO4 is highly expressed in the muscle and heart [127], and FoxO6 is expressed in the brain. The acetylation of FoxO reduces the attenuation of its transcriptional activity and mediates various biological functions [128]. Transcription factors of the FoxO family are known to be involved in the regulation of various genes related to antioxidant defense [129].

FoxO1 has been extensively studied and shown to play important roles in a variety of biological processes, including metabolism, proliferation, redox status, stress tolerance, inflammation, aging, and apoptosis [130]. The Sirt1/FoxO1 pathway has antioxidant effects because it increases the expression of SOD2 and CAT [130].

In renal tissue, FoxO1 activity was markedly reduced in diabetic kidneys, which was accompanied by extracellular matrix accumulation and oxidative stress [131,132].

However, FoxO3a regulates biological processes in cells, such as metabolism, cell proliferation and differentiation, oxidative stress tolerance, inflammation, senescence, and apoptosis [133,134]. Sirt1 inhibits FoxO3a-induced cell death by deacetylating FoxO3a and inducing autophagy and antioxidant effects by activating FoxO3a. In response to oxidative stress, Sirt1 regulates cellular responses by modulating the function of FoxO3 [133,135,136].

In the kidney, Sirt1 expression is markedly reduced with age, suppressing both autophagy and cell cycle arrest and leading to the accumulation of oxidative stress and apoptosis via acetylated FoxO3a, but it was restored in a study on Sirt1 downregulation in the aging kidney using a 12-month CR diet model [136]. Silencing of Sirt1 induces the overexpression of acetylated FoxO3a in tubular epithelial cells and exacerbates HG-induced oxidative stress [137].

Activation of Sirt1 alters the transcriptional activity of FoxO target genes, and activation of FoxO1 has antioxidant effects and reduces tissue damage and oxidative stress in diabetic kidneys and blood vessels. Activation of FoxO3a also attenuates oxidative stress and inhibits apoptosis in the kidney. The Sirt1/FOXO pathway may affect the kidney in terms of oxidative stress regulation and cell and tissue quality control.

#### 5.1.8. Sirt1 and HIF-1α

Sirt1 regulates the cellular response to hypoxia by deacetylating Nrf1α (HIF-1α) [138]. In CKD, renal hypoxia usually occurs during the course of renal disease progression, resulting in the expression of the oxygen sensor HIF-1α [139].

Activation of HIF-1 signaling also activates the vascular endothelial growth factor (VEGF) signaling pathway, which promotes angiogenesis [140]. The expression of other downstream genes, such as glucose transporter 1 (GLUT1) and erythropoietin (EPO), is also promoted [5]. Another function of HIF is to strengthen stress reduction and detoxification by inhibiting the production of ROS. HO-1 and SOD are well-known enzymes that are regulated by HIF [141,142].

Inactivation of Sirt1 leads to increased acetylation and the overexpression of HIF-1α, which is induced in DKD [138]. The overexpression of HIF-1α in the kidney promotes abnormal angiogenesis, causing glomerular hypertrophy and plasma leakage, which progresses to glomerulosclerosis and arterial hyalinization, leading to abnormal angiogenesis and fibrosis in the kidney [140]. In summary, the HIF-1α pathway is involved in oxidative stress and fibrosis of kidney tissues that occurs in various disease states, such as diabetes. However, Sirt1 activation may protect renal tissue via the regulation of the HIF pathway.

### 5.2. Sirt2

Sirt2 is localized in both the cytoplasm and nucleus and is widely expressed in various tissues, such as the brain, muscle, pancreas, liver, adipose tissue, and kidney. Sirt2 is a substrate for many histone and nonhistone proteins, such as histone H4 and tubulin [143]. Sirt2 is involved in genome integrity and cell function, including cell proliferation and differentiation, and energy metabolism, and Sirt2 activity has been associated with a variety of metabolic diseases [143]. Like other sirtuins, Sirt2 detects intracellular energy and has been shown to change its activity and expression in response to the energy status of the cell, becoming activated in low-energy conditions and becoming repressed in high-energy conditions [143].

Previous studies have reported that Sirt2 has antioxidant stress effects and maintains mitochondrial function in metabolism-related tissues; similar to Sirt1, Sirt2 regulates NF-ĸB, PGC-1α, HIF-1α, and FoxO transcription factors [144,145,146]. The deacetylation of FoxO by Sirt2 ameliorates oxidative stress via increased expression of the antioxidant enzyme SOD2 [147]. It has been reported that in the kidneys of CR mice, Sirt2 expression is upregulated and FoxO deacetylation is increased in response to calorie restriction and oxidative stress, resulting in decreased ROS at the cellular level [147].

A recent study showed that Sirt2 deacetylates Nrf2, resulting in a decrease in the total and nuclear levels of Nrf2 [148]. The results also revealed altered levels of total GSH and glutamate cysteine ligase (GCL), suggesting that Sirt2 may be an important regulator of this aspect of ARE [149]. In addition, Sirt2 activity mediates caspase-3 levels and influences oxidative stress, glucose 6-phosphate dehydrogenase (G6PD), phosphoglycerate mutase (PGAM2), and other metabolic enzymes [143]. Under oxidative stress conditions, Sirt2 has been shown to deacetylate and activate G6PD in the cytoplasm. G6PD is a key enzyme in the pentose phosphate pathway and regulates the oxidative stress response by producing NADPH and reducing GSH [150]. Sirt2 activates G6PD, which improves the redox status of cells and protects them from oxidative damage. Besides, under oxidative stress conditions, deacetylation and activation of PGAM2 by Sirt2 occurs, making cells more responsive to stress [151].

Sirt2 may be involved in mitochondrial regulation. Lemos et al. reported that Sirt2 is downregulated in a mouse model of obesity, which led to reduced extracellular signal-regulated kinase 1/2 (ERK1/2) activation and mitochondrial dysfunction [152]. Activation of Sirt2 increases the fusion-associated protein Mitofusin (Mfn) 2 and decreases mitochondria-associated dynamin-related protein 1 (Drp1), resulting in increased elongated mitochondria and improved mitochondrial function. Sirt2 also attenuates the downregulation of transcription factor A mitochondrial (TFAM), a major mitochondrial deoxyribonucleic acid (mtDNA)-related protein, resulting in increased mitochondrial mass [152]. However, we did not find any report that states that the regulation of Sirt2 with respect to the mitochondria has an impact on the kidney. These findings suggest that Sirt2 plays an important role in the regulation of oxidative stress responses. Sirt2 may be involved in protecting organisms from metabolic disorders through oxidative stress-dependent mechanisms. Sirt2 is thought to be involved in protecting organisms from metabolic disorders through oxidative stress-dependent mechanisms.

However, the role of Sirt2 in oxidative stress in the kidney has not been fully elucidated. Recent experiments using kidney tissue from Sirt2 knockout (KO) and Sirt2 transgenic (TG) mice have shown that the inflammation-related proteins CXCL2 and CCL2 were upregulated in both the mRNA and protein forms in the kidneys of lipopolysaccharide (LPS)-treated Sirt2 KO mice as compared to LPS-treated Sirt2 TG mice [153]. In addition, LPS-induced neutrophil and macrophage infiltration, acute tubular injury, and decreased renal function were suppressed in Sirt2 KO mice [153]. These results suggest that Sirt2 is associated with the expression of CXCL2 and CCL2 in the kidney and that overexpression of Sirt2 leads to the induction of renal injury. Furthermore, Sirt2 regulates the activation of the mitogen-activated protein kinase phosphatase 1(MKP-1), a dual-specificity protein phosphatase that transduces mitogen-activated protein kinase signaling in the kidney, and the MKP-1 expression was decreased in the kidney and tubular epithelial cells of Sirt2 TG mice [154]. Acetylation of MKP-1 was significantly increased in Sirt2-knockdown cells and was decreased in Sirt2-overexpressing cells in response to cisplatin stimulation [154]. In Sirt2 KO mice, cisplatin-induced renal injury, apoptosis, necrosis, and inflammation were ameliorated [154]. In contrast, Sirt2 TG mice displayed exacerbated cisplatin-induced renal injury, apoptosis, necrosis, and inflammation [154]. It has also been reported that suppression of Sirt2 activated by renal I/R by the administration of the Sirt2 inhibitor significantly reduced the number of apoptotic renal tubular cells and mitigated microstructural damage [147].

Thus, it has been suggested that overexpression of Sirt2 may be detrimental in the kidney. Therefore, whether Sirt2 expression has a useful function in the kidney is still uncertain and needs further investigation (Table 2).

### 5.3. Sirt3

Sirt3 is a major mitochondrial deacetylase and a major determinant of the mitochondrial acetylproteome, and its deacetylating activity has been implicated in mitochondrial biology and pathophysiological processes, such as redox status [155]. Much of the research on Sirt3 has focused on its role within mitochondria; the enzyme has been implicated in a variety of diseases, including cardiovascular disease, neurodegenerative disease, and even cancer, and has been reported to be involved in renal disease [156,157]. Sirt3 has also been recognized as an anti-aging molecule, and Sirt3 regulates mitochondrial metabolism and prolongs the lifespan of experimental animals [158]. High expression levels of Sirt3 have been associated with longevity in humans [159]. In addition, polymorphisms in the Sirt3 gene are known to correlate with decreased enzyme efficiency and the development of metabolic syndrome in humans, and there may be many links between Sirt3 and oxidative stress [160]. Based on a number of studies, Sirt3 is known to mediate the flow of mitochondrial oxidative pathways and thus regulate the production of ROS. Sirt3 has been shown to decrease ROS by deacetylating LKB1 and activating AMPK [161], thus regulating HIF-1α [162], and is reported to be an enzyme involved in the regeneration of the antioxidant GSH [163]; increased activity of Sirt3 is an intervention that promotes longevity [163]. The increased activity of Sirt3 partially explains the suppression of ROS by CR. Sirt3 also affects the production of ROS and influences the mitochondrial oxidative phosphorylation (OXPHOS) pathway by modulating enzymes involved in mitochondrial activation [164]. Human mitochondrial DNA (mtDNA) is more susceptible to oxidative damage than nuclear DNA because it encodes 13 proteins that are responsible for regulating mitochondrial respiration and oxygen degradation [165,166]. ROS produced by OXPHOS is generated in the cell. ROS generated by radiation and chemicals in the cell cause DNA damage and transform purine and pyrimidine bases into 8-oxo-7,8-dihydroguanine (8-oxoG) lesions. Importantly, Sirt3 has been shown to target human 8-oxoguanine-DNA glycosylase 1 (OGG1), which is the enzyme that repairs the DNA damage [167]. Sirt3 promotes OGG1 by physically binding to OGG1, preventing its degradation, and regulating the activation of OGG1 when DNA glycosylase is activated [167]. Sirt3 has been found to play an important role in repairing mtDNA and protecting cells from apoptosis under oxidative stress conditions by mediating the activation and replenishment of OGG1 [167]. In addition, mitochondria, despite their oval structure, undergo active and dynamic fission and fusion. Excessive fragmentation of mitochondria leads to mitochondrial dysfunction [168,169,170]. Mitochondria maintain their active and dynamic morphology by maintaining a series of fusion (Mfn-2; optic atrophy protein 1, OPA1) and fission cycles [171]. The fusion process protects the cell from mitochondrial crista structure and apoptosis [172]. Sirt3 can directly target OPA1. OPA1 is over-acetylated under stress conditions [172]. Acetylated OPA1 exhibits reduced GTPase enzymatic activity, which impairs its biological function. Sirt3 has been found to activate OPA1 by deacetylation, resulting in enhanced mitochondrial dynamics [173]. Taken together, these results suggest that Sirt3 plays an important role in protecting cells from stress.

Sirt3 also plays an important role in the kidney, especially in mitochondria-rich proximal and distal tubular cells. Mitochondrial hyperfragmentation and mitochondrial dysfunction in tubular cells have been identified as the central pathological features of renal tubular disorders [174]. It has been suggested that Sirt3 may play a role in maintaining mitochondrial energy homeostasis in proximal and distal tubular cells. Sirt3 has also been shown to mediate the deacetylation of enzymes involved in ROS reduction, leading to protection against oxidative stress-dependent pathogenesis and disease, including renal injury. For example, Sirt3 has been shown to activate isocitrate dehydrogenase (IDH2), SOD2, and CAT, which are key enzymes that reduce the intracellular burden of ROS [79,163,175]. Decreased Sirt3 activity has been found to increase oxidative stress by increasing the acetylation of IDH2 and SOD2 enzymes and decreasing the activation of mitochondrial antioxidants. In one study, mice lacking Sirt3 were compared to normal mice, and decreased oxygen consumption and increased ROS were observed in Sirt3 KO [176]. Furthermore, Sirt3 KO mice treated with cisplatin developed more severe AKI, impaired mitochondrial dynamics, and further accumulated oxidative stress [161]. Conversely, overexpression of Sirt3 attenuated I/R-induced mitochondrial damage, oxidative stress in renal tubular epithelial cells, tubular cell apoptosis, and accumulation of proinflammatory cytokines and oxidative stress in sepsis-induced AKI [174]. Sirt3 expression is decreased in the diabetic state, and modulation of Sirt3 protects tubular cells from mitochondrial oxidative stress. Previously, we found, in a rat model of type 2 diabetes, increased acetylation of SOD2 and IDH2 decreased their activity and induced mitochondrial oxidative stress, which in turn decreased the activity of Sirt3 due to a decrease in the mitochondrial NAD^+^/NADH ratio [177].

Taken together with the apparent role of Sirt3 in protecting cells from oxidative damage, these studies suggest that Sirt3 may be involved in multiple pathways related to the regulation of oxidative stress (Table 3).

### 5.4. Sirt4

The Sirt4 protein, localized in mitochondria, has the primary function of adenosine diphosphate (ADP) ribosylation. It is said to act on ADP-ribosylate glutamate dehydrogenase (GDH) using NAD, which inhibits GDH activity, limiting ATP production [178]. It also inhibits the pyruvate dehydrogenase complex (PDC) by removing lipoamide modifications from its E2 subunit [179]. Thus, Sirt4 has been reported to function as a cancer suppressor gene by inhibiting glutamine metabolism [180]. Sirt4 is also a very weak deacetylase and has been reported to decrease fatty acid oxidation by deacetylating methylglutaryl, hydroxymethylglutaryl, and 3-methylglutaryl groups from lysine residues and by inhibiting malonyl-coenzyme A (CoA) decarboxylase [181].

Sirt4 has been shown to be involved in the regulation of ROS production in mitochondria [182]. However, it is unclear whether and how Sirt4 affects the activation of antioxidant enzymes localized in the mitochondria. While reporting on chondrocytes, Dai et al. found that Sirt4 supplementation resulted in a significant reduction in ROS accumulation with the upregulation of antioxidant enzymes, such as SOD1, SOD2, and CAT [183]. Furthermore, they found that silencing Sirt4 in mildly degenerated chondrocytes suppressed the expression of antioxidant enzymes, resulting in an imbalance in ROS, which was also restored by Sirt4 administration [183]. These results suggest that Sirt4 may be involved in the homeostasis of oxidative stress in human osteoarthritis. However, experiments in mice investigating angiotensin II (Ang II)-induced cardiac hypertrophy revealed that overexpression of Sirt4 increased ROS, while Sirt4 KO increased and decreased ROS in the heart and mitochondria, respectively [184]. Similar results have been obtained in rat cardiomyocytes. More interestingly, a 2014 study found that obese patients with hepatic steatosis had significantly lower levels of circulating Sirt4 compared to healthy controls. The decrease in Sirt4 levels may be a response to a lack of exercise and a high-calorie diet (both factors that increase oxidative stress in many mammals), which decreases mitochondrial ROS [185].

In the kidney, information on the role of Sirt4 is still scarce. Sirt4 expression in the kidney has been observed and shown to be regulated by CR, preventing the onset and progression of age-related renal damage [133]. A previous report revealed a relationship between Sirt4 and the development of DKD: Sirt4 expression was decreased in a mouse model of type 2 diabetes, and HG-induced podocytes exhibited markedly suppressed proliferative capacity and increased apoptosis [186]. These data suggest that the overexpression of Sirt4 inhibits podocyte apoptosis and reduces podocyte damage under HG conditions. In addition, the inflammatory cytokines TNF-α, IL-1β, and IL-6 were increased in response to HG conditions but were markedly decreased in response to Sirt4 overexpression [186]. These results indicate that Sirt4 is a promising therapeutic target for DKD, but other hypotheses need to be evaluated in future studies.

Thus, the exact role of this sirtuin is still unknown. Expression of Sirt4 varies among different cell types, and further studies are needed to clarify the full capacity and action of Sirt4 in antioxidant responses and oxidative stress (Table 3).

### 5.5. Sirt5

Sirt5 is localized in the mitochondria and has potent lysine desuccinylation, demalonylation, deglutarylation, and very weak deacetylation activities. It targets cardiolipin and desuccinylates mitochondrial inner membrane electron transporters complex I, complex II, and ATP synthase [187]. Sirt5 targets cardiolipin and desuccinylates mitochondrial electron transporter complex I, complex II, and ATP synthase. Sirt5, identified by large-scale profiling studies, is also involved in cellular metabolism, detoxification, and energy production and mediates the apoptotic pathway [188], and Sirt5 knockdown cells have abnormalities in several pathways involved in mitochondrial function, such as oxidative phosphorylation, tricarboxylic acid cycle, and amino acid metabolism [189,190]. Furthermore, sirt5 detoxifies ammonia by deacetylating and activating carbamoyl phosphate synthase (CPS1), which catalyzes the first step in the urea cycle. Sirt5 KO mice exhibit elevated ammonia levels during fasting [191]. Conversely, mice overexpressing Sirt5 display increased activity of CPS1, which facilitates the conversion of ammonia to nontoxic urea [192]. Importantly, ammonia is known to induce the production of ROS and decrease the amount of GSH, which is an antioxidant [193]. Thus, it is suggested that Sirt5 is indirectly involved in the management of oxidative stress. Lin et al. found that Sirt5 enhances the activity of SOD1. In addition, co-expression of Sirt5 enhances the reduction of ROS by SOD1, revealing the post-translational regulation of SOD1 in lung tumor cells [194]. Sirt5 has also been shown to deacetylate cytochrome C, which is an essential component of the electron transport system, as well as enzymes that are thought to contribute to the inhibition of ROS [195]. Another study showed that levels of ROS were reduced in cells transfected with Sirt5, suggesting that Sirt5 inhibits the progression of oxidative stress conditions [196]. Taken together, these results suggest that Sirt5 plays an important role in the cellular response to oxidative stress.

Immunostaining of normal human kidneys shows that Sirt5 expression is increased in mitochondria-rich tubules, and depletion of Sirt5 reduces tubular cell energy metabolism, disrupts mitochondrial fission and fusion dynamics, increases mitochondrial swelling, and induces fragmentation and a decrease in mitochondrial respiration with an increase in mitophagy [197]. In fact, it has been reported that the restoration of reduced Sirt5 by cisplatin treatment maintains antioxidant capacity and mitochondrial homeostasis via Nrf2 regulation and inhibits apoptosis and ROS production [198]. In this report, Li et al. showed that a large number of apoptotic cells appeared in renal tissues in response to cisplatin exposure. They also showed for the first time that overexpression of Sirt5 partially attenuates the toxic effects of cisplatin by downregulating the levels of apoptosis [198], suggesting that Sirt5 plays an important role in the survival of renal cells under the cytotoxic pressure of cisplatin. However, it has been reported that Sirt5 KO mouse kidneys were protected against ischemic and cisplatin-mediated AKI [199]. In other words, it has been suggested that the loss of Sirt5 function may confer renoprotection. Moreover, a study comparing the I/R model and folate nephritis (FAN model) has been reported. In the mouse kidney, Sirt5 expression was increased in I/R but decreased in FAN [200]. Sirt5 KO mice were protected from I/R kidney injury, while the injury was mildly exacerbated in the FAN model. Sirt5 KO renal mitochondria showed reduced complex II activity, a central cause of injury in I/R. Loss of Sirt5 in mice in vivo alleviated IRI and exacerbated FAN in mice in vivo [200]. It has been suggested that loss of mitochondrial function may reduce acute ischemic injury but exacerbatie chronic nephrotoxic injury, and whether Sirt5 expression or loss of function actually protects the kidney is unknown at this stage and requires further study (Table 3).

### 5.6. Sirt6

Sirt6 is found in the nucleus and functions as an adenosine diphosphate (ADP)-ribosyltransferase and NAD+-dependent deacetylase [201]. Sirt6 deacetylates histone H3 at various lysine sites and regulates DNA repair, telomere maintenance, genome stability, and cellular senescence [201]. Sirt6 has been shown to be involved in the regulation of longevity. Overexpression of Sirt6 decreased NF-ĸB [201]. A previous report has implicated it in increasing serum levels of insulin-like growth factor (IGF)-1 and IGF binding protein 1 [202]. Sirt6 levels are increased in rats in response to CR conditions. Furthermore, overexpression of Sirt6 extends the lifespan of male mice, whereas Sirt6 KO mice exhibit early senescence and usually live for approximately 3 months [203]. Overexpression of Sirt6 further protects against high fat diet-induced metabolic abnormalities. It has also been reported that Sirt6 regulates metabolic changes via PGC-1a, p53, and FoxO1 signaling [204,205]. In other words, Sirt6 is considered to be an important metabolic sensor that links environmental signals to metabolic homeostasis and stress responses in mammals.

In response to oxidative stress, it has been hypothesized that Sirt6 activity is regulated via ROS-induced post-translational modifications [206]. Sirt6 expression was decreased in T2DM patients, as compared to nondiabetic patients, and was associated with the progression of atherosclerotic lesions [207]. A decreased expression of Sirt6 was also associated with increased oxidative stress and inflammation [207]. It has been reported that Sirt6 expression is involved in the expression of PARP1 in response to stress [208]. Overexpression of Sirt6 is associated with oxidative stress tolerance and regulates eNOS expression [209], reduces oxidative stress via AMPK and FoxO3a, and upregulates endogenous antioxidants, which have been shown to protect cardiomyocytes from I/R injury [210]. Similarly, Sirt6 has been shown to protect human mesenchymal stem cells (hMSCs) from oxidative stress through the coactivation of Nrf2 and HO-1 [211]. Conversely, in hMSCs lacking Sirt6, redox metabolism is dysregulated, and sensitivity to oxidative stress is increased. It has also been suggested that Sirt6 functions as a coactivator of Nrf2 via the interaction with the RNA polymerase II (RNAP II) complex and that the reduction of Sirt6 in hMSCs results in decreased activation of the RNAP II complex [211]. However, endothelial cells transcribed in human vasculature were unaffected by the loss of Sirt6, suggesting that differences in cell type may underlie these observed phenotypic differences [211]. Therefore, Sirt6 seems to be essential for redox and antioxidant homeostasis, but there may be some differences in its function in cells and organs.

In the kidney, Sirt6 is thought to be important in maintaining the function of podocytes and glomeruli [212]. Deletion of Sirt6 in mice induces podocyte damage and decreases the expression of slit diaphragm protein. Some studies have shown that Sirt6 plays a protective role in renal dysfunction. Silencing of Sirt6 in HK-2 renal epithelial cells promotes the secretion of cytokines, such as TNF-α and IL-6 [213]. Overexpression of Sirt6 attenuates LPS-induced cytotoxicity and apoptosis, and Sirt6 is involved in NF-ĸB signaling [213]. Moreover, extracellular Sirt6 suppresses NF-ĸB signaling and ERK1/2 expression, and overexpression of Sirt6 suppresses cisplatin-induced inflammation and oxidative stress-induced cytotoxicity and apoptosis [214]. Therefore, an increased expression of Sirt6 in renal cells may also inhibit the accumulation of inflammation and oxidative stress in renal components and blood vessels.

Overall, the expression and function of Sirt6 seem to overlap with anti-detoxification and antioxidant mechanisms and help to suppress ROS. It seems that the effect can be expected in the kidney as well. However, there are some differences in function in cells and organs, and the role and relevance of Sirt6 in oxidative stress and related diseases requires further study (Table 4).

### 5.7. Sirt7

Sirt7, the last member of the mammalian sirtuin family that has been identified, is also primarily localized in the nucleus [215].

Specifically, it is expressed in the nucleolus and is an NAD^+^-dependent deacetylase with a high selectivity for histone H3 acetylated lysine 18 (H3K18), which actively regulates ribosomal DNA (rDNA) transcription [216]. The expression levels of Sirt7 mRNA are different in all tissues, with higher expression levels in tissues with higher metabolic activity [217].

A previous study also revealed that Sirt7 is involved in the regulation of mitochondrial homeostasis via deacetylation of GA-binding protein subunit beta1 (GABPb1), one of the subunits of the complex involved in mitochondrial gene regulation [218]. However, in this study, it was unclear whether Sirt7 activity was a cause or a consequence of NAD^+^ availability [218], and further studies are needed to clarify the relationship between Sirt7 regulation of NAD^+^ and cellular energy levels. Sirt7 is slightly downregulated in H9c2 cells derived from the embryonic rat heart in response to oxidative stress induced by H_2_O_2_ [219].

Interestingly, Sirt7 KO mice experience premature physiological aging and appear older than their actual age [220]. This suggests that Sirt7 may be involved in aging and age-related diseases. During aging, Sirt7 moves from the nucleolus to the chromatin and cytoplasm, where it may suppress rDNA transcription [215]. However, it has been reported that Sirt7 KO mice are resistant to weight gain, fatty liver, and glucose intolerance induced by a high-fat diet, suggesting that the reduced expression of Sirt7 may have a protective effect [221].

While there are few reports in the kidney, there is one recent report stating that mice lacking Sirt7 are protected against cisplatin-induced AKI [222]; loss of Sirt7 suppresses the cisplatin-induced post inflammatory response, reduces the amount of injury, and ameliorates AKI via the suppression of NF-ĸB expression. Furthermore, it has been suggested that a loss of Sirt7 may reduce oxidative stress by converting TNF-α, which normally promotes the production of ROS, into an NADPH oxidase complex [222].

Sirt7 has also been reported to negatively affect the protein levels of HIF-1α in the kidney through a mechanism that is independent of prolyl hydroxylation and does not involve degradation in proteasomes or lysosomes [223]. However, the mechanism by which Sirt7 regulates HIF activity is independent of its deacetylation activity, suggesting that Sirt7, which is considered to be different from other sirtuins, may regulate HIF through protein–protein interactions rather than enzymatic activity, which may affect the kidney [223].

As Sirt7 is a relatively recent subject of study, our understanding of the substrates on which it acts and the mechanisms through which it plays various roles in the cell is still limited. However, it has been suggested that Sirt7 plays at least a minor role in regulating the cellular response to oxidative stress (Table 4).

### 5.8. Sirtuin Activators and Their Effects

Sirtuins have been reported to be activating substances. Resveratrol, an antioxidant found in grapes, is thought to activate sirtuins. There have been many reports on the effects of resveratrol administration. It has been reported that resveratrol administration reduces HG-stimulated renal mesangial cell fibrosis, glomerular epithelial cell damage, and podocyte damage via the AMPK, PGC-1α, and HIF-1α pathways [94,140,224,225,226]. Administration of resveratrol also induces deacetylation of NF-ĸB p65 and improves the survival of septic rats by inhibiting AKI-induced renal injury, and it was reported to attenuate renal hypertrophy and urinary albumin excretion in diabetic rats without affecting blood glucose levels [227,228]. With respect to oxidative stress, resveratrol has been shown to increase both eNOS gene expression and eNOS enzyme activity, increase FoxO1 activity and SOD activity in DKD conditions, exert nephroprotective effects, and modulate Nrf2-dependent antioxidant protein expression [229]. It has been shown to prevent H_2_O_2_-induced cell death, decrease cell proliferation, and inhibit aging. As explained before, Sirt7 is slightly downregulated in response to H_2_O_2_-induced oxidative stress in embryonal heart derived from the H9c2 cells of rats [219]. This study also found that this decrease was mitigated by pre-exposure to the resveratrol. This suggests that other oxidative stress-related activities of resveratrol may be due to Sirt7 activity, at least in part, and that further work needs to be done to determine the exact contributions of the other sirtuins to resveratrol’s successes.

Sirt1 activators other than resveratrol have also been reported to exert antioxidant and nephroprotective effects. Astragaloside IV (VAS-IV), an antioxidant, is thought to improve renal function and morphology by inhibiting mesangial cell activation via the Sirt1-NF-ĸB pathway, which may be useful in the treatment of glomerular diseases [230]. It has been reported that glycyrrhizic acid ameliorates HG-induced low expression and activity of AMPK, Sirt1, and SOD2 in renal tubular cells and improves DKD in a mouse model of type 2 diabetes by suppressing ROS and activating the AMPK/Sirt1 pathway [123]. Xu et al. reported that puerarin, a flavonoid, protects the kidney by upregulating the expression of Sirt1 and FoxO1 in the renal cortex [131].

There are three studies on SRT2104, SRT1720, and SRT3025, which are synthetic small molecule activators of Sirt1. SRT2104 and SRT1720 prolonged the lifespan and improved the overall health of mice fed a standard diet [231,232]. SRT2104 treatment was also found to increase SOD2 levels, suggesting that the activation of the Sirt1 small molecule activator enhances the antioxidant response to age-dependent and ROS-mediated mitochondrial dysfunction in mice [232]. Indeed, SRT3025 reduced proteinuria and ameliorated eGFR in a 5/6 nephrectomy rat model [233].

Furthermore, sodium-glucose cotransporter 2 (SGLT2) inhibitors, a new class of oral hypoglycemic agents, improve outcomes at both ends of CKD [234]. Recent studies have reported that SGLT2 inhibitors attenuate the progression of these changes through the Sirt1/PGC-1α axis [235]. SGLT2 inhibitors induce a fasting-like transcriptional paradigm involving activation of Sirt1. Sirt1/PGC-1α activation is a Sirt1/PGC-1α activation that acts as a master regulator of nutrition and cellular homeostasis, each of which promotes gluconeogenesis and ketogenesis, hallmarks of SGLT2 inhibitor treatment for nephroprotection [235].

In addition to Sirt1, Sirt3 activators are being studied. The activation of Sirt3 by honokiol, a small molecular weight polyphenol, reduced ROS in the cisplatin-induced AKI model by improving mitochondrial function [161]. Treatment with the AMPK agonist AICAR or the antioxidant agent acetyl-l-carnitine (ALCAR) restored Sirt3 expression and activity, improved renal function, and decreased tubular injury [174]. We confirmed and reported that apigenin, a flavonoid, was administered to diabetic model rats to activate Sirt3 by inhibiting NAD^+^-degrading enzyme CD38, and that mitochondrial oxidative stress and renal tubular cells injury of DKD were reduced by deacetylation of SOD2 and IDH2 [236]. The effects of Sirt2, Sirt4, Sirt5, Sirt6, and Sirt7 activators on the kidneys have been largely unreported.

In this way, in vitro and animal studies have shown that sirtuin activators, such as resveratrol, can have various renal protective effects. However, there have been few reports on human subjects.

Results from several clinical trials indicate that resveratrol may have cardioprotective effects, increasing antioxidant capacity, having a positive impact on circulatory function and anti-diabetic effects in humans [237]. Some of those reports have examined kidney function, but no significant results have been found [238,239,240]. In addition to resveratrol, we also studied the effects of red wine extracts containing catechin, epicatechin, quercetin, and anthocyanin, which have been suggested to activate Sirt1 in cell and animal experiments [241], and piceatannol, a hydroxylated analog of resveratrol, which is abundant in *Passiflora edulis* and has been shown to have Sirt1-inducing activities [242]. A small-scale clinical study was conducted on human subjects, and both of them reported some positive effects on glucose metabolism [241,242]. However, no beneficial effects on renal function were found in any of the studies.

Some studies have focused on NAD^+^, the substrate of sirtuins. The major pathway for the biosynthesis of NAD^+^, the substrate of sirtuins, includes the conversion of nicotinamide to nicotinamide mononucleotide (NMN) and the subsequent conversion of NMN to NAD^+^. The generation of NMN is an important rate-limiting factor in mammalian NAD^+^ biosynthesis. A recent report in humans showed that oral supplementation of NMN had a positive effect on humans by improving insulin sensitivity in muscle without major side effects [243]. Furthermore, it was recently reported that short-term administration of NMN to 8-week-old diabetic nephropathy mice for 2 weeks enhanced Sirt1 expression in podocytes and continued to inhibit albuminuria in a glucose-independent manner until 24 weeks of age [244]. Based on these recent results, there is a slight possibility that NMN, which has been shown to be safe and effective in humans, may inhibit the progression of renal damage by activating Sirt1 in the early stages of nephropathy. In addition, NMN is a precursor of NAD^+^ and may theoretically affect the activity of other NAD+-dependent sirtuins.

Taken together, these data suggest that sirtuin activators, directly or indirectly, play an important role in the prevention of renal oxidative damage through a variety of mechanisms (Table 2, Table 3 and Table 4).

However, whether sirtuin activators, such as resveratrol, have nephroprotective effects in humans is not certain from the results from clinical trials, and further studies are needed, taking into account, for example, the issues of dosage and duration of administration.

## 6. Considerations and Conclusions for Future Research

In summary, sirtuins play an important role in maintaining cellular homeostasis and cellular health, making them an ideal target for redox regulation studies. The maintenance of redox homeostasis is mainly related to transcriptional regulation through feedback mechanisms that function at various complex levels. As mentioned earlier, sirtuins regulate many of the important genes and molecules that are essential for redox homeostasis. Many mammalian sirtuins are thought to have bioprotective, antioxidant-promoting, and reactive oxygen species (ROS) inhibitory effects that affect the overall pathology of kidney tissue (glomeruli, blood vessels, and tubules). It has been suggested that increased oxidative stress due to the abnormal expression of sirtuins may lead to renal dysfunction. Regulation of sirtuins is thought to protect the constituent cells of the kidney. Overall, the apparent protective effect of sirtuins on oxidative stress is based on the theory that sirtuins act synergistically through a variety of mechanisms in order to enhance cellular homeostasis. Sirt1 has been the most studied, and its activation is likely to be nephroprotective at this stage. Sirt3 and Sirt6 have also been reported to have beneficial effects in recent years. However, considering that the loss of Sirt2, Sirt4, Sirt5, and Sirt7 has been reported to have a nephroprotective effect, it is important to control sirtuins “appropriately” from the concept of hormesis in the control of oxidative stress. Therefore, there is a major problem in developing selective sirtuin inhibitors/activators as a single therapeutic approach. Sirtuins are involved in a myriad of pathways, and the potential for beneficial effects of sirtuin activation in humans must be considered, as well as the potential for detrimental effects of over-activation. The development of either specific inhibitors or activators of sirtuins is fraught with risk. Indeed, despite the compilation of many experimental studies with promising results, there is currently little evidence of sirtuin-specific pharmacological activity in humans, and therefore the number of clinical trials and the results they highlight are very limited.

Therefore, sirtuins need further study as a general therapeutic target in the kidney. In particular, more specific clinical trials focusing on appropriate levels of sirtuin manipulation, relevant pathways, tissue and cell specificity, genetics, and epigenetics may be needed in the future. A possible compromise at this time would be to consider developing therapies that target the pathways regulated by sirtuins, rather than the sirtuins themselves.

## Figures and Tables

**Figure 1 antioxidants-10-01198-f001:**
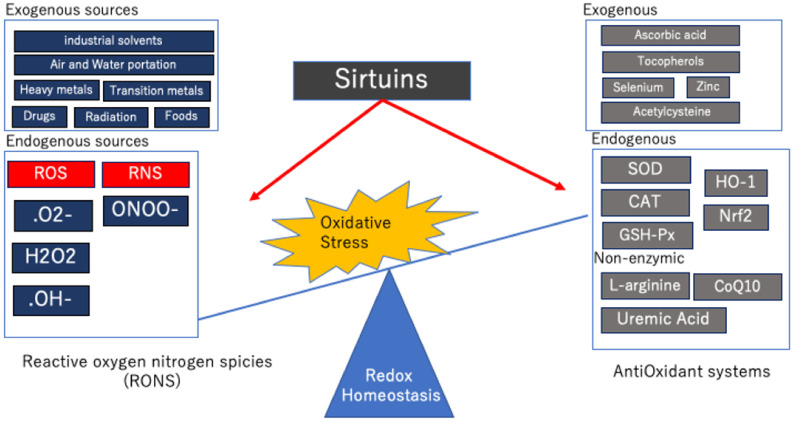
The imbalance between free radicals and antioxidant systems induces cell injury with consequent renal failure pathogenesis. Sirtuins are involved in processes and functions related to antioxidant and oxidative stress. Abbreviations: Reactive oxygen species: ROS, Reactive nitrogen species: RNS, Hydrogen peroxide: H_2_O_2_, Superoxide anion radicals: O_2_^•−^, Nitric oxide: NO, Nicotinamide adenine dinucleotide phosphate oxidase: NOX, Peroxynitrite: ONOO^−^, Super oxide dismutase: SOD, Catalase: CAT, Haem oxygenase 1: HO-1, and Glutathione peroxidase: GSH-Px.

**Table 1 antioxidants-10-01198-t001:** Mammals have seven sirtuins, each with a predominant subcellular localization and a different catalytic activity.

	Localization	Catalytic Activity
Sirt1	Nucleus and Cytoplasm	Deacetylation
Sirt2	Cytoplasm and Nucleus	Deacetylation
Sirt3	Mitochondria	Deacetylation
Sirt4	Mitochondria	ADP-ribosylation Deacetylase
Sirt5	Mitochondria	Desuccinylation Demalonylation Deglutarylation Deacetylation
Sirt6	Nucleus and Cytoplasm	Deacetylation ADP-ribosylation
Sirt7	Nucleus and Cytoplasm	Deacetylation

**Table 2 antioxidants-10-01198-t002:** Molecules targeted by sirt1 and sirt2 for regulation of the oxidative stress and the activator of sirt1 and sirt2.

	Target Molecules	Kidney Model	Activetor
Sirt1	NF-ĸB SOD2 CAT Nrf2 p53 eNOS p66Shc PGC-1α LKB1/AMPK FoxO1 FoxO3a HIF-1α	Diabetic kidney Aging kidney HG cultured tubular epithelial cell HG cultured podocyte Human embryonic kidney cells	CR Resveratrol VAS-IV Glycyrrhizic acid Puerarin SRT1720 SRT2104 SRT3025 SGLT2 inhibitors
Sirt2	NF-ĸB SOD2 PGC-1α HIF-1α Nrf2 G6PD PGAM2 ERK1 DRP1 Mfn2 TFAM MKP-1	Renal Ischemia/Reperfusion model LPS nephropathy Cisplatin-induced Acute kidney injury	CR

Abbreviations: Nuclear factor kappa β: NF-ĸB, Super oxide dismutase: SOD, Catalase: CAT, Nuclear factor erythroid 2-related factor 2: Nrf2, Endothelial nitric oxide synthase: eNOS, The adaptor protein 66 kDa Src homology 2 domain-containing protein: p66shc, Peroxisome proliferator-activated receptor-γ coactivator-1α: PGC-1α, Liver kinase B1: LKB1, AMP-activated protein kinase: AMPK, Forkhead Box O:FoxO, Hypoxia-inducible factor: HIF, Glucose 6-phosphate dehydrogenase: G6PD, Phosphoglycerate mutase: PGAM2, Extracellular signal-regulated kinase: ERK, Dynamin-related protein: Drp, Mitofusin: Mfn, Transcription factor A mitochondrial: TFAM, Mitogen-activated protein kinase phosphatase: MKP-1.

**Table 3 antioxidants-10-01198-t003:** Molecules targeted by sirt3, sirt4, and sirt5 for regulation of the oxidative stress and the activator of sirt3, sirt4, and sirt5.

	Target Molecules	Kidney Model	Activator
Sirt3	OGG1 OPA1 SOD2 IDH2 CAT AMPK HIF-1α	Cisplatin-induced Acute kidney injury Renal Ischemia/Reperfusion model Sepsis-induced Acute kidney injury Diabetic kidney	Honokiol Apigenin AICAR ALCAR
Sirt4	GDH PDC malonylCoA decarboxylase SOD CAT	Diabetic kidney Aging kidney HG cultured podocyte	CR
Sirt5	Mitochondrial Complex 1 Mitochondrial Complex 2 ATP synthase CPS1 SOD1 Cytochrome C Nrf2	Cisplatin-induced Acute kidney injury Renal Ischemia/Reperfusion model Folate nephritis	Fasting

Abbreviations: 8-oxoguanine-DNA glycosylase 1: OGG1, Super oxide dismutase: SOD, Catalase: CAT, Optic atrophy protein 1: OPA1, Isocitrate dehydrogenase: IDH2, AMP-activated protein kinase: AMPK, Forkhead Box O: FoxO, hypoxia-inducible factor: HIF, ADP-ribosylate glutamate dehydrogenase: GDH, Pyruvate dehydrogenase complex: PDC, Carbamoyl phosphate synthase: CPS1, Nuclear factor erythroid 2-related factor 2: Nrf2.

**Table 4 antioxidants-10-01198-t004:** Molecules targeted by sirt6 and sirt7 for regulation of the oxidative stress and the activator of sirt6 and sirt7.

	Target Molecules	Kidney Model	Activator
Sirt6	NF-ĸB PGC-1α p53 FoxO1 FoxO3a AMPK eNOS Nrf2	LPS nephropathy Cisplatin-induced Acute kidney injury	CR
Sirt7	GABPb1 NF-ĸB HIF-1α	Cisplatin-induced Acute kidney injury	CR Resveratrol

Abbreviations: Nuclear factor kappa β: NF-ĸB, Peroxisome proliferator-activated receptor-γ coactivator-1α: PGC-1α, Forkhead Box O: FoxO, AMP-activated protein kinase: AMPK, Endothelial nitric oxide synthase: eNOS, Nuclear factor erythroid 2-related factor 2: Nrf2, GA-binding protein subunit beta1: GABPb1, Hypoxia-inducible factor: HIF.

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
