# Peer review of "Sirtuins and Renal Oxidative Stress"

_antioxidants, 2021, doi:10.3390/antiox10081198_

Round 1

Reviewer 1 Report

In numerous experimental models, redox active compounds such as antioxidants induce hormetic dose responses that are not only common but display endpoints of biomedical and clinical relevance. These hormetic responses are mediated via the activation of nuclear factor erythroid- derived 2 (Nrf2) antioxidant response elements (AREs) and, as such, are characteristically biphasic, well integrated, concentration/dose dependent, and specific with regard to the targeted cell type and the temporal profile of response. In experimental disease models, the polyphenol-induced hormetic activation of Nrf2 was shown to effectively reduce the occurrence and severity of a wide range of human-related pathologies, including Parkinson's disease, Alzheimer's disease, stroke, age-related ocular damage, chemically induced brain damage, and renal nephropathy, amongst others, while also enhancing stem cell proliferation.

Interestingly, the mechanistic profile of SFN or HD is similar to that of numerous other hormetic agents, indicating that activation of the Nrf2/ARE pathway is probably a central, integrative, and underlying mechanism of hormesis itself. The Nrf2/ARE pathway provides an explanation for how large numbers of agents that both display hormetic dose responses and activate Nrf2 can function to limit age-related damage, the progression of numerous disease processes, and chemical- and radiation- induced toxicities. The hormetic dose response is generally recognized as a reliable feature of the dose response for oxygen free radicals and their redox regulated transcriptional factors as well as antioxidant compounds and appears to have an important impact on brain pathophysiology and stress resistance mechanisms to oxidative and inflammatory insult and neurodegenerative damage.

This is an interesting paper. The study is well-conceived and well-executed. This reviewer is satisfied with the significance of this study, the care in which the study was performed, and the implications of the results for human health.

However, although the results presented are convincing, the work raises some concerns which will need to be addressed. The questions posed are of extremely high interest, but the paper does not give adequate definitive information, therefore pending addressing some major question is possible to accept for publication.

Minor concerns:

1. Preconditioning signal leading to cellular protection through Hormesis is an important redox dependent aging-associated to free radicals species accumulation, inflammatory responses involved in oxidant-induced cell damage and also reparative/cytoprotective mechanisms. This aspect should be highlighted in the discussion and references properly added (See Calabrese et al., 2010, Antiox. Redox Signal 13,1763; Siracusa et al., Antioxidants 2020, 9(9):E824)

2. Given the relationship between antioxidant compounds, redox status and the vitagene network and its possible biological relevance in neuroprotection, Authors while interpetrating results should discuss appropriately this aspect and make proper connection with emerging principles of hormesis ; line 138 please correct: Antioxidant defense s mechanisms …

Author Response

Thank you very much for your medical review of my submission. I have added the concerns you raised. 

1.
Thank you very much. I have added a new unit 2-3. on hormesis and oxidative stress, and discussed the concept of hormesis and specific examples from the references you provided. I have also added a discussion in the conclusion.

2.
Thank you, I have added vitagenes to the section on anti oxidant defense mechanisms and neuroprotection against oxidative stress. I have also corrected the typos that you pointed out.

Thank you very much for your kind attention.

Reviewer 2 Report

This review article shows the relationship between sirtuins and oxidative stress in various renal disease. Authors emphasize that sirtuins play an important role in maintaining cellular homeostasis, which means novel target for treatment of kidney disease. Therefore, this review article has worthy to publication for understanding the role of sirtuins.

I have minor comments.
1. Page 4, line 184~187: This paragraph describes the abbreviation. So, this sentence should delete or shift to the proper position (insert to legend for figure 1).
2. What is the most important one among Sirt1~Sirt7? After reading this review article, many researchers feel that Sirtuins have various beneficial effects on kidney diseases. So, what are the authors’ opinions about the development of drugs targeting sirtuins to treat kidney diseases? The additional description might help to understand drug development.
3. Are there any reports about clinical trials using sirtuin activators or inhibitors in kidney disease?

Author Response

Thank you very much for your medical review of my submission. I have added the concerns you raised.

1.
Thank you for your suggestion. As you suggested, I have inserted to legend for figure 1.

2. and 3.
Thank you for raising the question. We believe that Sirt1 has the best effect on the kidneys to date, but we believe that other sirtuins also have hidden nephroprotective effects that are not yet clear, and we cannot say for sure which one is important at this stage.
However, there are few reports of nephroprotective effects of any of the sirtuins in human clinical trials, so we believe that there are still some problems. This year's report on the administration of the latest NAD precursor to humans has been added because of its potential to activate sirtuins.
I have added these discussions to 5.8. Sirtuin activators and their effects and 6. Considerations and Conclusions for Future Research.

Thank you very much for your kind attention.
